

# Taehwa Research Forest: A receptor site for severe pollution events in Korea during 2016

John T. Sullivan[1], Thomas J. McGee[1], Ryan M. Stauffer[1,2], Anne M. Thompson[1], Andrew Weinheimer[3], Christoph Knote[4], Scott Janz[1], Armin Wisthaler[5,6], Russell Long[7], James Szykman[7,8], Jinsoo Park[9], Youngjae Lee[9], Saewung Kim[10], Daun Jeong[10], Dianne Sanchez[10], Laurence Twigg[1,11], Grant Sumnicht[1,11] Travis Knepp[8,12] and Jason R. Schroeder[13]

[1]Atmospheric Chemistry and Dynamics Laboratory, NASA Goddard Space Flight Center, Greenbelt, MD, 20771, USA
[2]Universities Space Research Association, Columbia, MD, 21046, USA
[3]National Center for Atmospheric Research, Boulder, CO, 80305, USA
[4]Meteorologisches Institut, Ludwig-Maximilians-Universität München, München, Germany
[5]Department of Chemistry, University of Oslo, Oslo, Norway
[6] Institute for Ion Physics and Applied Physics, University of Innsbruck, Innsbruck, Austria
[7] US EPA/Office of Research and Development/National Exposure Research Lab, Research Triangle Park, NC, 27711, USA
[8]NASA Langley Research Center, Hampton, VA, 2368, USA
[9]National Institute of Environmental Research, Incheon, South Korea
[10]Department of Earth System Science, University of California, Irvine, Irvine, CA, USA
[11]Science Systems and Applications, Inc., Lanham, MD, 20706, USA
[12]Science Systems and Applications, Inc., Hampton, VA, 23666, USA
[13]California Air Resources Board, Sacramento, CA, 95814, USA

**Correspondence:** John T. Sullivan (john.t.sullivan@nasa.gov)

**Abstract.** During the May-June 2016 International Cooperative Air Quality Field Study in Korea (KORUS-AQ), light synoptic meteorological forcing facilitated Seoul metropolitan pollution outflow to reach the remote Taehwa Research Forest (TRF) site and cause regulatory exceedances on 24 days. Two of these severe pollution events are thoroughly examined. The first, occurring on 17 May 2016, tracks transboundary pollution transport exiting eastern China and the Yellow Sea, traversing the Seoul Metropolitan Area (SMA), and then reaching TRF in the afternoon hours with severely polluted conditions. This case study indicates that although outflow from China and the Yellow Sea were elevated with respect to chemically unperturbed conditions, the regulatory exceedance at TRF was directly linked in time, space, and altitude to urban Seoul emissions. The second case studied, occurring on 09 June 2016, reveals that increased levels of biogenic emissions, in combination with amplified urban emissions, were associated with severe levels of pollutions and a regulatory exceedance at TRF. The case studies are assessed with multiple aircraft, model (photochemical and meteorological) simulations, in-situ chemical sampling, and extensive ground-based profiling at TRF. These observations clearly identify TRF and the surrounding rural communities as receptor sites for severe pollution events associated with Seoul outflow, which will result in long-term negative effects to both human health and agriculture in the affected areas.



## 1 Introduction

The spatiotemporal characteristics of ozone ($O_3$), nitrogen dioxide ($NO_2$) and other urban pollutants have been monitored at the ground level within the Seoul Metropolitan Area (SMA) and throughout the Republic of Korea (commonly referred to as South Korea) for several decades [Seo et al., 2014]. Although trans-boundary transport events from other countries (*e.g.* China) have been demonstrated [Choi et al., 2014], several studies have clearly illustrated the impacts of domestic pollutants on rural receptor sites downwind of the SMA [Kim et al., 2012; Jeon et al., 2014]. Furthermore, negative effects associated with poor air quality in South Korea [Ghim et al., 2007] have been connected with increased mortality rates [Lee et al., 2000], and reduction in agricultural yields [Wang and Mauzerall (2004)].

Recent work with high-resolution satellite records from Duncan et al., 2016 indicate a decreasing trend in tropospheric $NO_2$ column throughout the SMA from 2005 to 2014 that has been attributed to regulatory (*e.g.* vehicular) controls or a transition to low/zero emissions vehicles [Wang et al., 2015]. However, during this same time period, the observed trend has increased near the petrochemical and industrial regions to the west and southwest of the SMA. Therefore, due to the complexities and heterogeneity of emissions within the SMA [Vellingiri et al., 2015], it is important to assess chemical gradients that may occur among sites with varying distances from emission sources [Ryu et al., 2013; Iqbal et al., 2014, Jeon et al., 2014; Lee et al., 2014], including sites that may be rural pollution receptor sites.

While a dense network of ground level observations over South Korea exists to monitor pollution, information regarding $O_3$ and its precursors (*e.g.* $NO_2$; volatile organic compounds (VOCs), including biogenic VOCs (BVOCs)) above the surface has been sparse. Because of the lack of chemical profiles, it is difficult to characterize $O_3$ and other pollutants aloft and quantify the impacts of vertical mixing down to the ground level. Analysis of vertical profiles of $O_3$ at the Olympic Park (OLY, 37.5232°N, 127.1260°E, 26m ASL) site within the SMA [Kim et al., 2007] indicate photochemical $O_3$ production in the afternoon hours corresponding mainly to local precursor advection from upwind regions (the western part of SMA). More recent findings describe the highest $O_3$ events occurring at rural sites 30 km (Taehwa Forest (TRF, 37.3123°N, 127.3106°E, 160m ASL; Kim et al., 2013) and 100 km (Chuncheon, 37.881°N, 127.676°E; Jeon et al., 2014) from the SMA. Both of those studies linked the high pollution events to mobile (vehicular) source emissions in the presence of natural biogenic emissions (*i.e.* advection of SMA emissions away from their origins and into a high-BVOC environment of increased $O_3$ production efficiency).

To further investigate the vertical distribution of pollutants impacting South Korea, the United States (U.S.) National Aeronautics and Space Administration (NASA) and the Korean Ministry of the Environment/Korean National Institute of Environmental Research (NIER) conducted an international cooperative field experiment with sampling at both the TRF and OLY sites, entitled the Korea-U.S. Air Quality (KORUS-AQ, https://espo.nasa.gov/home/korus-aq/content/KORUS-AQ) study. The KORUS-AQ observation period was specifically chosen to target local photochemical pollution (which peaks in May-June) rather than pollution transport which tends to be greatest in March-April. Differences in daily average (24-hr) $NO_x$ (NO + $NO_2$, Figure 1c) and maximum daily hourly $O_3$ (Figure 1d) during the KORUS-AQ study are presented from 10 May



to 11 June 2016. NO$_x$ (which is predominantly NO$_2$ during daytime) can rapidly form O$_3$ in the presence of VOCs and favorable meteorology. The two largest sources within the South Korea NO$_x$ emissions inventory are mobile/vehicular emissions (41.7%) and road transport and "other" mobile sources (20.0%) [Lee et al., 2011].

The prevalence of vehicular emissions is apparent in the urban environment (Figure 1c) with NO$_x$ amounts at OLY frequently an order of magnitude greater than those at rural TRF. Herman et al., 2018 has further shown during the KORUS-AQ study that the difference between TRF (also referred to as Taehwa Mountain) and OLY in columnar NO$_2$ can be as much as 3.0 DU (Dobson Units). The concentrations of surface O$_3$ at TRF during stagnant flow regimes far exceeded those at OLY (Figure 1d) on several days during the study, indicating O$_3$ formation within the SMA is VOC limited, due to the overabundance of NOx. However O$_3$ formation can rapidly increase as the distance from NOx sources increases. Recent
chemical simulations performed during the KORUS-AQ period by Miyazaki et al., 2018 also reveal that observed boundary layer O$_3$ can be as much as 30 ppbv different with dynamic and stagnant flow regimes. Supporting work at TRF has also shown that fast oxidation rates (Kim et al., 2018) and overall oxidation capacity (Jeong et al., 2018) at TRF can exacerbate severe pollution events.

      The differences in daily maximum hourly O$_3$ between the TRF and OLY sites vary from day to day, but the sites had
mean and 1-hr maximum concentrations of 86.1 ± 21.9 ppbv and 80.4 ± 17.5 ppbv, respectively. The South Korean national standard for O$_3$ is 100 ppbv (parts-per-billion by volume) for a 1-hr average and 60 ppbv for an 8-hr average (http://eng.me.go.kr/eng/web/index.do?menuId=253). Both sites exceeded the 8-hr standard 24 times between 10 May and 10 June 2016. However, TRF exceeded the national 1-hr standard 11 times, whereas OLY exceeded on only 3 days. These daily and diurnally varying O$_3$ amounts during the KORUS-AQ study at TRF are described with back-trajectories (2.1), synoptic
meteorology (2.2) and balloon-borne profiles from TRF (3.1.1). Since the evolution of plume composition is a critical component in understanding severe O$_3$ exceedances, two contrasting case studies (17 May (3.2) and 09 June (3.3)) of the 11 TRF 1-hr exceedances are examined with aircraft and ground-based measurements including O$_3$ lidar and ceilometer backscatter profiles. These are complemented by a photochemical box model used to calculate net O$_3$ production and are used distinguish urban/industrial emissions from one of mixed urban and biogenic origins (4).

## 2   Meteorological Analyses

### 2.1   Back-trajectories

      To understand variations in the air mass history as it is advected towards TRF, 4D (time, height, latitude, longitude) back-trajectories were simulated for every day during the KORUS-AQ study (Figure 2). The back-trajectory calculations were
performed using the Lagrangian FLEXible PARTicle dispersion model (FLEXPART, http://flexpart.eu; Brioude et al., 2013), driven by the WRF (Weather Research and Forecasting) model meteorology at 3 km spatial resolution. For this simulation, thousands of "air parcels" were released at 15:00 KST (Korean Standard Time; UTC –9 hrs), and their spatial and vertical



(Figure 2b) trajectory locations were followed back in time for 6 hrs. Chaotic processes like turbulence or convection were applied in a stochastic manner to each parcel. At hourly intervals the concentration of parcels in each cell of a regular grid was calculated, thus providing the best statistical estimate of the air mass location and altitude prior to reaching TRF.

Red trajectory paths in Figure 2 are used to identify air masses that were associated with the 11 $O_3$ exceedance days at TRF (based on the South Korean National Standard of 1-hr standard of 100 ppbv in Figure 1d). Two of these days, 17 May and 09 June (detailed in section 4), are shown in yellow and orange, respectively. Conversely, days where the daily 1-hr $O_3$ at TRF did not exceed 100 ppbv are shown in blue. Days that exceeded the 1-hr standard had generally shorter trajectory paths, indicating they were associated with more stagnant conditions and weaker synoptic forcing. The altitudes of exceedance day air masses (Figure 2b) were also much closer to the surface (thus closer to ground level anthropogenic and biogenic emissions) as compared to non-exceedance days. Exceedance days at TRF were also associated with air masses that flowed through or near the southern portion of the SMA, where they were more likely to interact with local anthropogenic and industrial emission sources [Iqbal et al., 2014].

### 2.2 Geopotential Height Anomalies

The Modern-Era Retrospective analysis for Research and Applications Version 2 (MERRA-2, https://gmao.gsfc.nasa.gov/reanalysis/MERRA-2/; Gelaro et al., 2017) 500 hPa geopotential heights and anomalies (1981-2010 base period) are presented for 17 May (Figure 3a) and 9 June 2016 (Figure 3b) at 21:00 KST. An upper level ridge exists on 17 May near northeastern China as shown by positive geopotential height anomalies. Under a surface high-pressure and light synoptic forcing, winds were generally light (*e.g.* 1-2 ms$^{-1}$ observed at TRF) and westerly in the morning and afternoon, changing to calm winds at TRF after 14:00 KST. Based on the back-trajectory (Figure 2), the light westerly winds in the morning transported pollutants from the western portion of the SMA to the eastern portion. As the flow became more quiescent in the afternoon, local emissions were pooled in the south-eastern portion of the SMA and then continued directly into the rural forested area near TRF. This synoptic system also ushered in warm (24° C and 27° C at 15:00 KST at TRF and OLY, respectively) and cloud-free conditions throughout the day. With this synoptic meteorology, unimpeded solar radiation, and ample $O_3$ precursor emissions from SMA, TRF reached a maximum hourly $O_3$ value of nearly 120 ppbv (Figure 1d) at 17:00 KST, well above the 100 ppbv South Korean national standard.

In contrast, on 09 June a weak 500 hPa ridge existed over South Korea, the Yellow Sea, and eastern China with a +50 m height anomaly. Under this multi-day weak synoptic forcing, the back trajectory indicates recirculation of the air mass throughout the Korean peninsula, rather than extended zonal transport. The air mass appears to have tracked near Seoul on the previous day, followed by recirculation back to the densely forested region south and west of TRF. Light and northwesterly surface winds occurred in the morning and afternoon (e.g. 2-3 ms$^{-1}$ at TRF), decreasing and becoming more southerly in the late afternoon (e.g. less than 0.5-1 ms$^{-1}$ at TRF after 12:00 KST). Throughout the day, the light north-westerly transported pollutants from the northwest to the southeast within the SMA. As the flow reduced in the afternoon and became more southerly, local fresh (as well as aged/recirculated) emissions were pooled throughout the southern portion of the SMA,



yielding adequate time to interact with the rural forested area near TRF. This system was also associated with much warmer (27° C and 31° C at 15:00 KST at TRF and OLY, respectively) conditions than 17 May, favoring increased emissions of BVOCs, such as isoprene [Kim et al., 2014]. With weakly forced synoptic meteorology, TRF reached a maximum hourly $O_3$ value of nearly 110 ppbv, exceeding the 1-hr 100 ppbv South Korean national standard.

## 3 Case Studies of Pollution Transport to TRF

### 3.1 Methods

Vertical profiles of key atmospheric chemical constituents measured via aircraft and from ground based platforms (Table 1) during two representative case studies are used to better understand trans-boundary and local transport effects from urban regions to the rural landscape. To assess the transport and evolution of urban emissions impacting TRF, airborne

measurements of $O_3$, $NO_2$, CO (carbon monoxide), $SO_2$ (sulfur dioxide), isoprene ($C_5H_8$), and toluene ($C_6H_5$-$CH_3$) were collected. Downwind plume chemistry is also further investigated and fingerprinted using the onboard DC-8 observations and an explicitly constrained 0-D photochemical box model simulation. Constituent profiles at TRF were also measured using ground-based instrumentation such as $O_3$ lidar (Sullivan et al., 2016; 2017), electrochemical cell (ECC) $O_3$-sondes, and aerosol backscatter. Surface observations of $O_3$, $NO_2$, toluene, and isoprene at TRF are also presented.

The instruments onboard the NASA DC-8 and Hanseo University King Air provide accurate, fast response measurements of trace gases and can be used in conjunction with aircraft-based remote sensing instruments in some cases to extend the characterization of pollution events. The Geostationary Trace gas and Aerosol Sensor Optimization (GeoTASO) airborne instrument was onboard the NASA B-200 King Air performing push broom raster sample routes to complement flights. Measurements of backscattered solar radiation are used to determine slant column (slcol) amounts of $NO_2$ at 250 m ×

250 m spatial resolution, providing a quantitative spatial distribution of $NO_2$ throughout the SMA.

### 3.1.1 Ozonesonde Profiles at TRF

A total of 34 $O_3$-sondes were released from TRF throughout KORUS-AQ from 10 May to 12 June. Afternoon soundings (13:30 to 16:30 KST) of $O_3$ (top panel, Figure 4) and temperature (bottom panel, Figure 4) illustrate day-to-day variability in the first 3km ASL. There were large disparities in boundary layer $O_3$ throughout the campaign period. From 10-

16 May, concentrations were mostly between 70-80 ppbv, which were associated with cooler temperatures and higher synoptic wind speeds. However, by the early afternoon of 17 May, a stagnant high-pressure system located over the Yellow Sea (c.f. Figure 3a) introduced a warmer air mass, calmer winds, and clearer skies. Through most of the campaign until 3 June, a similar meteorological setup persisted, providing favorable conditions for rapid $O_3$ production to more than 120 ppbv. On 4 and 6 June, intermittent shower activity events limited $O_3$ production. However, by 9-10 June another high-pressure system (c.f.

Figure 3b) approached the region, increasing temperatures, suppressing wind speeds, and fostering $O_3$ build-up. Two case



studies (17 May; 9 June, black boxes) are presented in this section and Figure 4 emphasizes the regularity of these enhanced levels of pollution.

### 3.2 Pollution Event: 17 May 2016

#### 3.2.1 Aircraft Analyses: Hanseo University King Air

The positioning of the large-scale ridge displayed in Figure 3a was favorable for transport of Chinese industrial (e.g. CO; $SO_2$) and urban emissions to the Korean peninsula [Lee et al., 2007]. To assess potential trans-boundary transport from other East Asian megacities, trajectories are shown for a full 48-hrs prior to reaching TRF at 17 May at 15:00 KST. The 48-hr back-trajectory (top panel, Figure 5) indicates the air mass was transported over the eastern China province of Shangdong, the Yellow Sea, and SMA prior to reaching TRF. Pollution entering South Korea via this pathway was measured by NIER

instruments (Table 1) onboard the Hanseo University King Air aircraft during the morning of 17 May 2016 (Figure 5).

The Hanseo University King Air conducted a sampling pattern that included a near ground level approach in Seoul and westbound leg out towards the Yellow Sea (green panel, Figure 5), a southbound leg directly over the Yellow Sea at two altitudes (orange panel, Figure 5), a returning eastbound leg (magenta panel, Figure 5) and then finally a northbound returning leg towards Seoul (cyan panel, Figure 5). During this pattern, measurements of $O_3$, $NO_2$, CO and $SO_2$ capture long-range

pollution transport across the Yellow Sea. Eastern China is densely populated with coal-fired power plants, which are strong emitters of $NO_x$, $SO_2$ and particulate matter [Zhao et al., 2008]. Carbon monoxide and $SO_2$ are longer lived species with lifetimes on the order of 1-2 months [Miyakazi et al., 2012] and 1-2 days [He et al., 2012], respectively, and are used to support the interpretation of trans-boundary pollution transport. During the KORUS-AQ study period, Huang et al., 2012 has further used CO to evaluate chemical transport models and assess transboundary impacts on the Korean peninsula.

During the low level approach to Seoul near 08:55 KST (denoted with dashed black line, Figure 5) chemical sampling of low $O_3$ (20-40 ppbv), high $NO_2$ (20-40 ppbv), high CO concentrations (500-700 ppbv) and high $SO_2$ (6-10 ppbv) are observed in the first 500 m ASL. These indicate morning urban emissions (and subsequent $O_3$

titration) and because of the proximity to the surface level, these pollutants are largely associated with local SMA vehicular and industrial morning emissions. As the aircraft moves westward at 1000 m ASL, it samples a much cleaner air mass, but

reaches a plume of polluted air near 09:20 KST, associated with increased concentrations of $O_3$ (to 90 ppbv), CO (to 500-600 ppbv) and $SO_2$ (to 6-8 ppbv). In conjunction with the 48-hr back trajectory, this is likely the outflow of aged industrial emission from Eastern China [Zhao et al., 2008] that has been transported over the Yellow Sea. Similar concentrations of these species are observed during the southbound leg at 500 m ASL, indicating that the vertical distribution of pollutants is relatively well mixed in the polluted air mass from 500-1000m ASL.

During the eastbound leg at 1500 m ASL, the aircraft samples mostly clean air (similar to the outbound leg with $O_3$ near 60 ppbv), with the exception of a pollution plume at 10:55 KST. As the aircraft returns to the Korean peninsula, it observes relatively cleaner conditions until it approaches the southern portion of the SMA near 11:30 KST. As it descends



within the SMA, it encounters a rapid increase in concentrations of all species, with pronounced increases in concentrations of NO$_2$ (to 15-25 ppbv), CO (to 600-800 ppbv), and SO$_2$ (to 4-6 ppbv).

In summary, the westbound in-situ observations indicate transport of polluted air across the Yellow Sea towards South Korea. The largest chemical perturbations (e.g. 15-40 ppbv in NO$_2$; 6-10 ppbv in SO$_2$, 200-300 ppbv in CO) during the flight
pattern were spatially correlated with local emission sources during the initial ground level approach and the final transect nearing the SMA. The southbound transect indicated on this day that the background level of O$_3$ is near 60 ppbv and a 20-30 ppbv enhancement in O$_3$ is observed at 500m over the water. However, the increases in NO$_2$ are chemically responsible for rapid O$_3$ production, which were observed in near negligible amounts during the trans-boundary transects.

### 3.2.2 Aircraft Analyses: NASA DC-8

Chemical observations from the NASA DC-8 throughout the SMA from 15:05 to 15:40 KST on 17 May 2016 are presented (Figure 6). The aircraft route (Figure 6a) begins in the southern region at 2.1 km ASL, moving northwards towards TRF and descending to near 1.7 km ASL. The aircraft maintains this altitude on a northward track and turns westward and descends to near 1.0 km ASL towards Seoul to perform a near ground level pass near the Seoul/Incheon airport. The aircraft continues at 1.0 km ASL, overpasses TRF near 0.5 km ASL, descends to nearly 0.3 km ASL, and then quickly ascends out of
the boundary layer. The constituents shown from the flight path are O$_3$ (Figure 7b), NO$_2$ (Figure 7c), isoprene (Figure 7d), toluene (Figure 7e) and O$_3$ production (P(O$_3$), Figure 7f).

Toluene, a reactive aromatic and industrial VOC, is a useful tracer for urban anthropogenic emissions because it is a highly reactive O$_3$ precursor with a chemical lifetime on the order of a day. Toluene is a dominant VOC throughout the SMA and contributes to nearly 60.7% of the total VOC emissions [Lee et al., 2011]. Isoprene, a BVOC and derivative of
photosynthesis, is largely associated with deciduous trees (e.g. oak, which accounts for 85% of broadleaf trees in South Korea [Lim et al., 2011]). Isoprene can enhance photochemical O$_3$ production, is emitted almost entirely during the daytime, can form additional oxidative byproducts, and is released more abundantly with increased temperatures. Previous results from Kim et al., 2014 indicate that isoprene accounts for most of the midday hydroxyl radical (OH) reactivity (11-15 KST) at TRF and can rapidly increase O$_3$ production rates.

Within the SMA and below 0.5 km ASL, there is a significant chemical perturbation as compared to the free tropospheric concentrations of NO$_2$ (40-50 ppbv, Figure 6c) and toluene (5-7 ppbv, Figure 6c). These increases both lead to increases in modeled P(O$_3$) (10-20 ppbv/hr, Figure 6f). Isoprene is mostly less than 0.3 ppbv during this sampling. The concentrations of NO$_2$ have increased by 10-20 ppbv since the afternoon Hanseo University King Air sampling (Figure 5), indicating a persistent reservoir of reactive nitrogen coming from SMA throughout the day. Ozone remains mostly between
70-80 ppbv (Figure 6b) during the initial descent into Seoul, indicating emissions are fresh enough that rapid O$_3$ production has not occurred yet (which is further corroborated with the P(O$_3$) model output).

As the aircraft moves towards TRF, it samples various spatial chemical gradients. For example, NO$_2$ and toluene concentrations decrease to near 20-30 ppbv and near 2-3 ppbv, respectively. An increase in isoprene concentrations to 0.5-0.8



ppbv corresponds to the forested region southeast of Seoul. Directly over TRF, $O_3$ concentrations are increased as compared to those near Seoul by 5-10 ppbv, indicating that downwind $O_3$ production has increased with diluted levels of $NO_x$. The aircraft passes TRF and continues its eastward descent to 0.3 km ASL; $O_3$ concentrations markedly increase to between 110-125 ppbv and $P(O_3)$ rapidly increases to between 20-32 ppbv/hr. Although the aircraft samples nearly negligible concentrations

of isoprene and $NO_2$, toluene concentrations have increased and are comparable to those sampled near Seoul (3-4 ppbv). This is a strong indicator of an aged urban air mass containing highly reactive $O_3$ precursors impacting rural sites. In summary, trans-boundary pollution transport was observed via the Hanseo University King Air (Figure 5) on 17 May, but locally emitted $O_3$ precursors can be confidently attributed as a catalyst for the highest levels of boundary layer $O_3$ production observed near TRF.

### 3.2.3  Ground-Based Observations at TRF

To fingerprint and quantify the transported pollution reaching TRF, diurnally resolved observations are presented in Figure 7 for the entirety of 17 May.

*08:00 to 12:00 KST:*  Ceilometer observations (Figure 7a) and GSFC $O_3$ lidar observations (Figure 7b), both
containing an aerosol mixing height retrieval (black line), indicate residual layers and vertical stratification. Above the boundary layer between 1000-1700 m ASL, enhanced aerosol backscatter and concentrations of $O_3$ near 70-80 ppbv are similar in concentration and altitude to the trans-boundary pollution observed during the morning Hanseo University aircraft flight (Figure 5). Atmospheric layering also exists below 850 m ASL with a distinct high $O_3$ (100-120 ppbv) region near 600 m ASL and low $O_3$ (20-40 ppbv)/high aerosol backscatter from 600-850 m ASL. Although $NO_2$ concentrations are enhanced from 5
to 15 ppbv during this segment, near 09:00 KST, residual layer entrainment appears to occur as surface $O_3$ (Figure 7c) abruptly increases from 40 ppbv to 60 ppbv.  Increases in nitrate aerosol, either through $N_2O_5$ hydrolysis or morning residual layer entrainment, within the SMA during the KORUS-AQ study have been further examined by Kim *et al.,* 2018.  Near 11:30 KST, toluene (Figure 7d) levels increase to peak values for the day, near 8-9 ppbv, indicating the transport of urban industrial emissions (including the increased $NO_2$) to the site is well underway.

*12:00 to 19:00 KST:*  The boundary layer is convectively well-mixed to 1200 m ASL with $O_3$ concentrations near 85-95 ppbv, similar to the in-situ surface monitor. This plume has similar chemical composition as the plume observed "downwind"/east of TRF in the DC-8 observations, indicating $O_3$ production continued as it moved away from the SMA. $O_3$ concentrations begin to decrease (by 25 ppbv) near 14:00 KST, in conjunction with a 25 ppbv increase in $NO_2$, conserving total odd oxygen ($O_x$). Near 15:20 KST, the DC-8 overflew TRF and sampled comparable concentrations of $O_3$ and $NO_2$ as
the lidar, sonde and surface measurements. As emissions continue to photochemically process and advect over the TRF site, there is a rapid increase in $O_3$, well-mixed throughout the boundary layer, at TRF between 16:00 and 17:00 KST. Ozone concentrations increase by 60 ppbv and $NO_2$ decreases by 15 ppbv, indicating total $O_x$ is not conserved and a passing plume of urban emissions has arrived at TRF. Isoprene (Figure 7d) does not show a rapid change throughout the day; however it is



near peak concentration during this time. Concentrations of $O_3$ above 125 ppbv persist until 18:30 KST, although the mixing height decreases rapidly during this time. The in-situ observations indicate concentrations were above 100 ppbv until 18:30, indicating polluted conditions persisted well into the evening at the surface and even longer aloft. This later afternoon buildup and transport is also identified with results from [Lennartson et al., 2018] during the KORUS-AQ study, which indicate TRF

had consistently higher aerosol optical depth (AOD) values near 0.4–0.6 in the morning, decreasing throughout the day, and eventually rising again in the early evening at 15:00–16:00 KST.

   *19:00 – 23:00 KST:*  After sunset (near 19:00 KST), increases in aerosol backscatter and $O_3$ near 1500 m ASL are observed, indicating a stable residual layer persisted into the evening, trapping pollutants at TRF. As surface $O_3$ quickly decayed to near 20 ppbv after 19:00 KST, it corresponded to increases in $NO_2$ from near 15 to 20-30 ppbv and toluene from 2

to 4 ppbv, which corroborate the incoming pollution plume quantified with the DC-8 observations (Figure 6c,e). This indicates TRF was continuously perturbed by local urban emissions into the evening hours and this has likely affected the next day's chemical composition (e.g. Figure 4 indicates $O_3$ at TRF on 18 May exceeded >100 ppbv). Although the combined suite of aircraft (Figure 5) and lidar/ceilometer observations (Figure 7a/b) suggest transboundary $O_3$ and pollutants reached TRF site, the lidar and in-situ observations clearly indicate domestic anthropogenic emissions were the dominant source of the $O_3$

exceedance at TRF.

### 3.3   Pollution Event: 9 June 2016

#### 3.3.1   Aircraft Analyses: NASA DC-8

Chemical observations were made with the NASA DC-8 instruments in a similar pattern to those on 17 May 2016 throughout the SMA from 15:20 to 15:55 KST on 09 June 2016 (Figure 8a). However, the aircraft remained at lower altitudes during the

flight pattern prior to the initial upper level pass of TRF. This resulted in low-level sampling of the forested region and recirculated air mass. This aged air mass was associated with increased concentrations of $O_3$ to over 120 ppbv (Figure 8b), low concentrations of $NO_2$ (less than 5 ppbv; Figure 8c), variable concentrations of toluene (between 5-10 ppbv; Figure 8e), variable concentrations of isoprene (between 0.1-0.7 ppbv; Figure 8d), and $P(O_3)$ values between 20-35 ppbv/hr (Figure 8f).

   As the aircraft ascended out of the boundary layer prior to reaching Seoul, $O_3$ remained above 100 ppbv to near 2000

m ASL. Mixing heights are 500m deeper than on 17 May, presumable which as a result of warmer temperatures and greater convective mixing. As the DC-8 ascended out of the overpass south of Seoul, $NO_2$ concentrations reached 20-30 ppbv, toluene reached 3-6 ppbv and isoprene exceeded 1.2 ppbv (nearing the peak concentration measured via the DC-8 during the entire campaign). This air mass was also associated with lower values of $O_3$ and $P(O_3)$ as compared to the forest plume south of TRF, between 75-90 ppbv and 10-20 ppbv/hr, respectively, indicating the anthropogenic and biogenic emissions were still fresh.

Similar to 17 May, as the aircraft leaves urban Seoul and approaches TRF there is a strong spatial gradient in nearly all chemical constituents. At TRF, $O_3$ and $P(O_3)$ increase to over 120 ppbv and 30 ppbv/hr, respectively, while $NO_2$ decreases to 5-10 ppbv, indicating $O_3$ production was rapidly occurring and impacting rural sites downwind of Seoul.




### 3.3.2 Aircraft Analyses: NASA B-200

On 09 June 2016, the NASA B-200 performed a morning and afternoon raster (Figure 9) of the greater SMA from 12:00-14:00 KST and 14:00 – 16:00 KST, respectively. This yields a unique view of the concentrations and chemical transport of $NO_2$ throughout the SMA during the afternoon hours. During the 12:00-14:00 KST sampling, there is a clear maximum in $NO_2$ slant columns in the south and west of Seoul. Afterwards, the 14:00-16:00 KST measurements show the advection of $NO_2$ (and presumably other urban pollutants) eastward and southward throughout the SMA. During the 14:00-16:00 flight, large $NO_2$ column amounts extend to the southeastern portion of the SMA, with enhanced levels of $NO_2$ reaching the edge of TRF.

### 3.3.3 Ground-Based Observations at TRF

To fingerprint and quantify the chemical transport reaching TRF, diurnally resolved observations are presented in Figure 7 for the entirety of 9 June.

*08:00 to 12:00 KST:* Similar to 17 May, vertical profiles of aerosol backscatter and $O_3$ (Figure 10a;b) throughout 09 June indicate residual layering of the atmosphere in the morning hours. There exist descending layers of aerosols above the residual layer associated with the recirculation of the air mass from the previous day. Near 09:00 KST entrainment of the residual $O_3$ corresponds to an abrupt surface $O_3$ increase from 20 ppbv to 55 ppbv (Figure 10c). TRF is impacted with increased surface $NO_2$ (Figure 10c) towards 30 ppbv and toluene (Figure 10d) towards 4 ppbv during this time. Warmer temperatures on 09 June compared to 17 May lead to higher daytime isoprene concentrations (0.5 ppbv versus 0.3 ppbv).

*12:00 to 19:00 KST:* Boundary layer concentrations of $O_3$ and aerosols are well mixed with steady growth in $O_3$ from 12:00 until 16:30 KST, when a rapid influx of $O_3$ and aerosol occurs. This occurs after a significant positive perturbation in isoprene, while preceding an abrupt increase in $NO_2$ and toluene. The $O_3$ peak closely corresponds to the $O_3$ sampled via the DC-8 south and west of TRF, and in conjunction with the back-trajectory (Figure 2), it appears $O_3$ was likely advected through TRF during this time.

*19:00 to 23:00 KST:* As solar radiation declines, aloft concentrations of $O_3$ near 100 ppbv mixing to 2000 m ASL are observed, indicating a stable residual layer persisted into the evening and likely impacted the next day's $O_3$ composition (e.g. Figure 4 indicates $O_3$ at TRF on 10 June exceeded >100 ppbv). As surface $O_3$ quickly decayed to near 20 ppbv after 19:00 KST, it corresponded to increases in $NO_2$ from near 15 to 20-30 ppbv and toluene from 8 to 10 ppbv, which points to the incoming pollution plume captured with the GeoTASO observations (bottom panel, Figure 9), indicating TRF was impacted by regional urban emissions into the evening hours. In conjunction with the aircraft, lidar, and surface in situ observations, this case study emphasizes the role of domestically produced emissions (both biogenic and anthropogenic) in perturbing the chemical composition downwind of Seoul.



## 4 Case Studies in the Context of the Entire KORUS-AQ Study

### 4.1 Aircraft Observation and P(O₃)

Although two case studies are presented to illustrate the contrasting types of pollution influences at TRF, it is important to

assess how representative events were for the entire KORUS-AQ study. To do this, the chemical observations of isoprene (Figure 11, left panel) and toluene (Figure 11, right panel) on-board the NASA DC-8 are presented for all remaining afternoon flights (exact dates can be found here: http://www-air.larc.nasa.gov/missions/korus-aq/). Data was used when the DC-8 aircraft was below 1.5 km ASL within 1-degree latitude and longitude of the TRF site. They are compared with the photochemical box model results in order to illustrate the relative contribution of VOCs and BVOCs on $O_3$ production during the campaign.

In both panels of Figure 11, the case studies chosen (17 May – red, 09 June – blue) appear to be accurate representations of "typical" pollution events seen at TRF as recorded by all other days (black dots).

     For isoprene, 17 May had concentrations centered around 0.5 ppbv and were associated with $P(O_3)$ values between 5-15 ppbv/hr. However, 09 June concentrations of isoprene were well over 0.5 ppbv, extending to over 2.0 ppbv. These were associated with $P(O_3)$ values in excess of 20 ppbv/hr and nearing 35 ppbv/hr, indicating that biogenic emissions contributed

more to net $O_3$ production on this day than on 17 May. The 09 June case yielded nearly the highest isoprene-driven $O_3$ production rates during the campaign.

     For toluene, 17 May appears to have several focused regions of toluene nearing 7 ppbv mostly associated with $P(O_3)$ at or below 15 ppbv/hr. However, there is a subset of toluene with concentrations between 3-5 ppbv that are associated with $P(O_3)$ rates between 20-35 ppbv/hr. This is indicative of the aged urban plume associated with $O_3$ photolysis reaching the area

east of TRF. On 09 June, concentrations of toluene are nearly all below 5 ppbv and are grouped much closer together. This is indicative of a much more well-mixed air mass (which was also suggested with the Geo-TASO observations). Although 9 June was associated with a larger contribution of isoprene driven $O_3$ production, it had a similar concentration of toluene as 17 May, indicating that both contrasting high pollution events (and $P(O_3)$) were associated with high levels of urban pollutants.

### 4.2 GSFC O₃ Lidar Derived Campaign Average

NASA GSFC $O_3$ lidar profiles at TRF during all flight days during the KORUS-AQ campaign (Figure 12) can be used to derive a diurnal campaign average (c.f. Sullivan et al., 2015b). The early morning low $O_3$ feature is prominent in the composite figure, as well as the aloft residual $O_3$ concentrations we have linked to trans-boundary transport between 65-75 ppbv above 500 m ASL. As solar radiation and convective mixing increase in the late morning hours (after 11:00 KST), surface concentrations of $O_3$ better correlate with concentrations measured aloft from the lidar. In the afternoon hours, $O_3$ increases at

a rate of 5 ppbv hr$^{-1}$ between 500-1500m ASL and peak $O_3$ is observed between 17:00-18:00 KST. The difference between aloft and surface concentrations in the evening can be linked to a decoupling of the surface layer and rapid depletion and titration of surface $O_3$ from local $NO_x$ emissions. This process isolates the polluted aloft residual layer during night-time hours



that will potentially affect downwind locations on the following morning. This confirms that during the May-June time period, the late day peak $O_3$ occurrence is a persistent feature and is more enhanced than at the surface.

## 5 Conclusions

As part of the KORUS-AQ study, nearly continuous chemical measurements at the surface and within the first 3km ASL have quantified several of the key pollution features (e.g. residual layer $O_3$ entrainment and late-day $O_3$ maxima) responsible for $O_3$ exceedances observed at a rural site downwind of Seoul. The combination of aircraft (NASA DC-8, Hanseo University King Air, and NASA B-200), in-situ (surface level and balloon-borne) and remotely sensed ($O_3$ lidar, ceilometer) measurements, in conjunction with photochemical model simulations, have produced significant findings about the origins of pollution reaching

TRF that appear to be representative of other rural sites in South Korea. Two detailed case studies have been presented, which indicate a mixture of urban/anthropogenic emissions and biogenic emissions impacting TRF. These case studies are also characteristic of the entire study (c.f. Figure 12), suggesting late day $O_3$ increases occur frequently at TRF, and that rural sites in this region may be experiencing long term negative effects of $O_3$. Because threshold health effects, mortality rates, and crop yield analyses have been historically calculated using only surface measurements [Lee et al., 2000; Kim et al., 2004; Wang

and Mauzerall (2004), Ghim et al., 2007], Figure 12 also indicates that these analyses may be underestimating the extent of the negative impacts of high-$O_3$ at TRF and its surrounding rural areas.

These results clearly demonstrate that Korea is subject to highly (aged) polluted air masses that cross the Yellow Sea but are exacerbated by domestic pollution produced near SMA. This emphasizes a reevaluation of domestic emission controls, in particular reactive aromatics such as toluene (*e.g.* toluene not only contributes to the $P(O_3)$ shown herein, but also contributes

to 9% of modeled secondary organic aerosol over SMA [Nault et al., 2018]). Organic aerosol formation has also been recently investigated during the KORUS-AQ study period to estimate relationships between in situ observations and satellite derived products (e.g. formaldehyde, (Liao et al., 2019)). These findings are in line with the detailed Rapid Science Synthesis Report (espo.nasa.gov/sites/default/files/documents/KORUS-AQ-ENG.pdf) that provides findings from the KORUS-AQ study which are intended to be useful for policy makers as they develop air quality mitigation strategies and continue to identify specific

emission sources that should be targeted for reduction. Direct observations of free tropospheric $O_3$ were rarely observed below 60 ppbv during the KORUS-AQ study period, indicating that the baseline conditions of which South Korean national regulatory standards are predicated on, are trending (Cooper et al., 2016) towards a regime where they are increasingly unattainable. In order to further assess trans-boundary pollution, emission sources and plume evolution, there has been an international effort to launch the Geostationary Environmental Monitoring Spectrometer (GEMS) to provide hourly

measurements of key pollutants (e.g. $O_3$, $NO_2$, $SO_2$ and particulate matter) over the Korean peninsula and the Asia-Pacific region. The KORUS-AQ analyses offer an exemplary synergistic approach on how to collect the statistics required by the regulatory agencies of Korea to improve air quality in both urban and rural settings.



*Acknowledgements*: The KORUS-AQ study could not have been completed without the leadership and shared partnership between Korea's National Institute of Environmental Research (NIER) and the United States National Aeronautics and Space Administration (NASA). This research was supported by multiple appointments to the NASA/USRA Postdoctoral Program at the Goddard Space Flight Center Atmospheric Chemistry and Dynamics Laboratory and Langley Research Center. The

authors gratefully acknowledge support provided by the NASA Tropospheric Chemistry Program and the Tropospheric Ozone Lidar Network (TOLNet). The authors would like to thank the entire KORUS-AQ science team for thoughtful discussions. We would also like to thank the pilots/captains and crew of the Hanseo University King Air, NASA B-200, and NASA DC-8. PTR-ToF-MS measurements aboard the NASA DC-8 during KORUS-AQ were supported by the Austrian Federal Ministry for Transport, Innovation and Technology (BMVIT) through the Austrian Space Applications Programme (ASAP) of the

Austrian Research Promotion Agency (FFG). The PTR-MS instrument team (P. Eichler, L. Kaser, T. Mikoviny, M. Müller) is acknowledged for their support with field work and data processing. Furthermore, this work could not have been completed without the heritage of observations and existing infrastructure at the Taehwa Research Forest site, operated by the College of Agricultur and Life Sciences at Seoul National University.

*Data Availability:* Unless otherwise noted, all data used in this study is available in the KORUS-AQ data archive (http://www-

air.larc.nasa.gov/missions/korus-aq/).

*Author contributions.* JTS, TJM, RMS, AMT, LT, GS, designed and executed field measurements for collecting surface $O_3$, $O_3$ lidar, and ozonesondes at TRF; AnW provided the airborne $O_3$/$NO_2$ data; ArW provided the airborne PTR-To-MS data; CK ran the FLEXPART analysis; SJ provided the GeoTASO data; RL, JS, and TK provided surface and ceilometer data; SK, DJ, and DS provided the PTR-ToF-MS data and logistical support; JRS provide the photochemical modeling analysis; JP and

YL provided additional surface measurements at TRF and OLY; JTS prepared the original manuscript, and all other authors contributed in editing the manuscript.

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





5  **Table 1.** Measurement species divided by each platform used throughout the case studies.

| Quantity | Method | Reference |
|---|---|---|
| **Hanseo University King Air** | | |
| $O_3$ | UV Absorption, Teledyne T400 | Kim, S.-Y. et al., 2013 |
| $NO_2$ | CAPS, Teledyne T500U | |
| CO | UV-Fluorimetry, AeroLazer (AL5002) | |
| $SO_2$ | UV-Fluorimetry, Thermo 43i | |
| **NASA DC-8** | | |
| $O_3$ ;$NO_2$ | Chemiluminescence | Weinheimer A. J., 2006 |
| Isoprene; Toluene | Proton-Transfer-Reaction Time-of-Flight Mass Spectrometer (PTR-TOF-MS) | M. Müller et al., 2014 |
| $P(O_3)$ | 0-D Photochemical Box Model | Schroeder et al 2016; (MCM v3.3, http://mcm.leeds.ac.uk/MCM/) |
| **NASA B-200 King-Air** | | |
| $NO_2$ slant column | GeoTASO | Nowlan et al., 2015 |
| **Taehwa Research Forest (TRF)** | | |
| $O_3$ | Lidar, UV differential absorption | Sullivan et al., 2014; 2015a; |
| $O_3$; Temp; RH; Winds | Electro-chemical cell, balloon-borne | Thompson et al., 2007; Witte et al., 2017; Thompson et al., 2019 (EOR) |
| $O_3$ | UV absorption, Thermo 49i | www.thermofisher.com/49i-Datasheet.pdf |
| $NO_2$ | CAPS, Teledyne T500U | www.cfpub.epa.gov |
| Atm. Backscatter | Vaisaila CL-51 | www.cfpub.epa.gov |
| Isoprene; Toluene | Proton-transfer-reaction mass spec (PTR-ToF-MS) | Kim et al., 2010 |



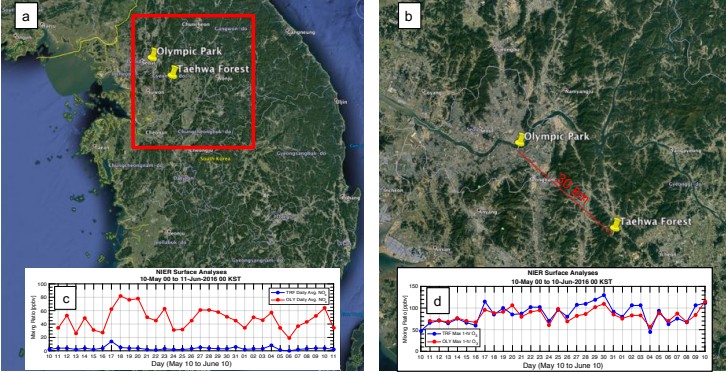

Figure 1. a) Map of the Korean Peninsula, b) inset view of the Olympic Park and Taehwa Research Forest sites, c) daily average $NO_x$ and d) maximum hourly $O_3$ at the TRF and OLY sites.



**Figure 2.** Daily 6-hr FLEXPART back-trajectory a) spatial and b) vertical locations for all parcels initialized at 15:00 KST at

TRF throughout the KORUS-AQ campaign. The two case studies are 17 May (yellow) and 09 June (orange).



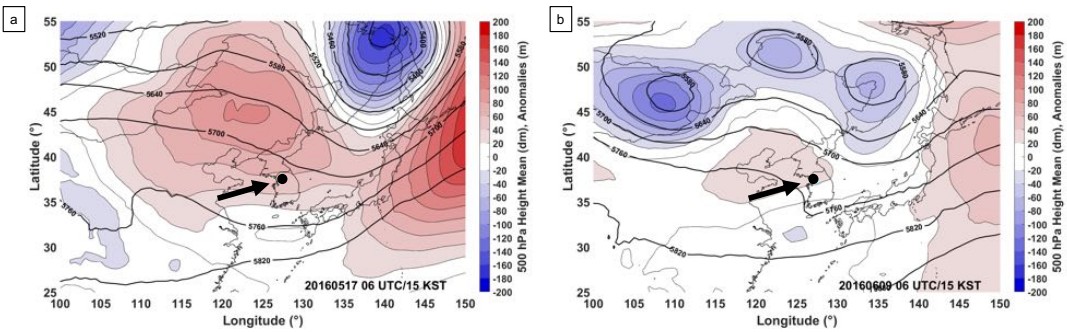

**Figure 3.** MERRA-2 500 hPa geopotential heights and anomalies for a) 17 May and b) 09 June 2016 at 21:00 KST (TRF

marked in black dot/arrow).



Figure 4. Afternoon soundings (13:30 to 16:30 KST) of O₃ (top panel, from ECC O₃-sondes) and temperature (bottom panel, from concurrent radiosondes) illustrate day-to-day variability at TRF. The two case studies (17 May and 09 June) are highlighted with black boxes.





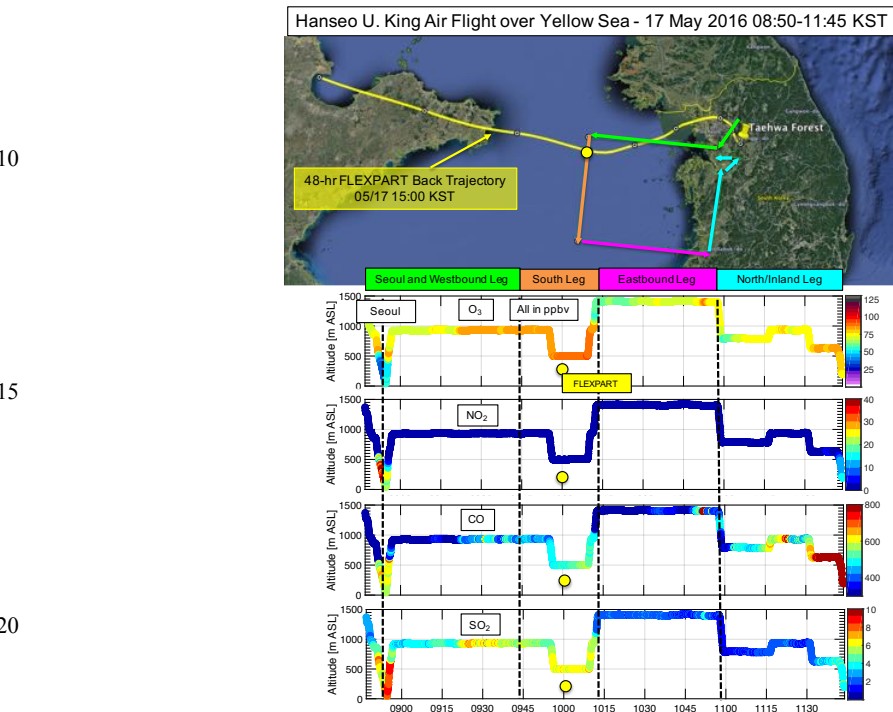

**Figure 5.** Hanseo University King Air Flight on 17 May 2016 to sample $O_3$, $NO_2$, CO, and $SO_2$ over South Korea and the Yellow Sea from 08:50-11:45 KST. The 48-hr FLEX-PART back trajectory, which was initialized at 15:00 KST on 17 May 2016, is also shown.





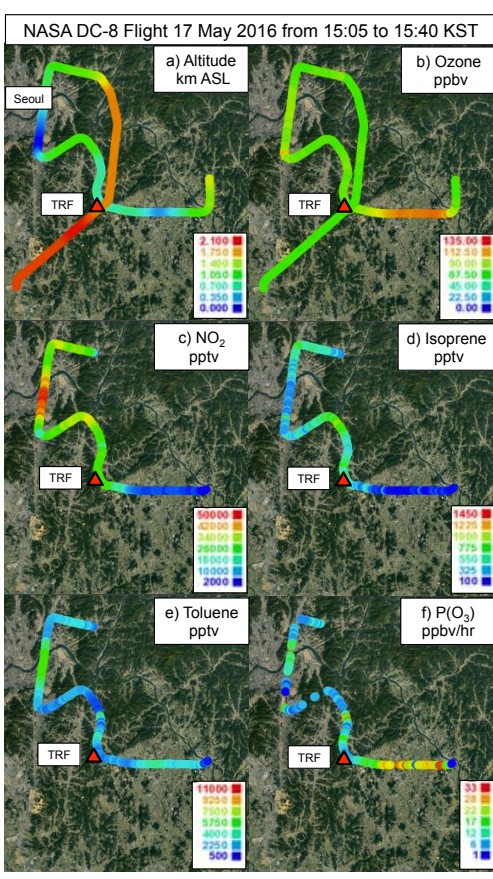

**Figure 6.** NASA DC-8 measurements of a) altitude, b) $O_3$, c) $NO_2$, d) isoprene, e) toluene and f) modeled $P(O_3)$ from the

25  afternoon science flight on 17 May 2016 from 15:05 to 15:40 KST. Seoul and TRF are denoted.




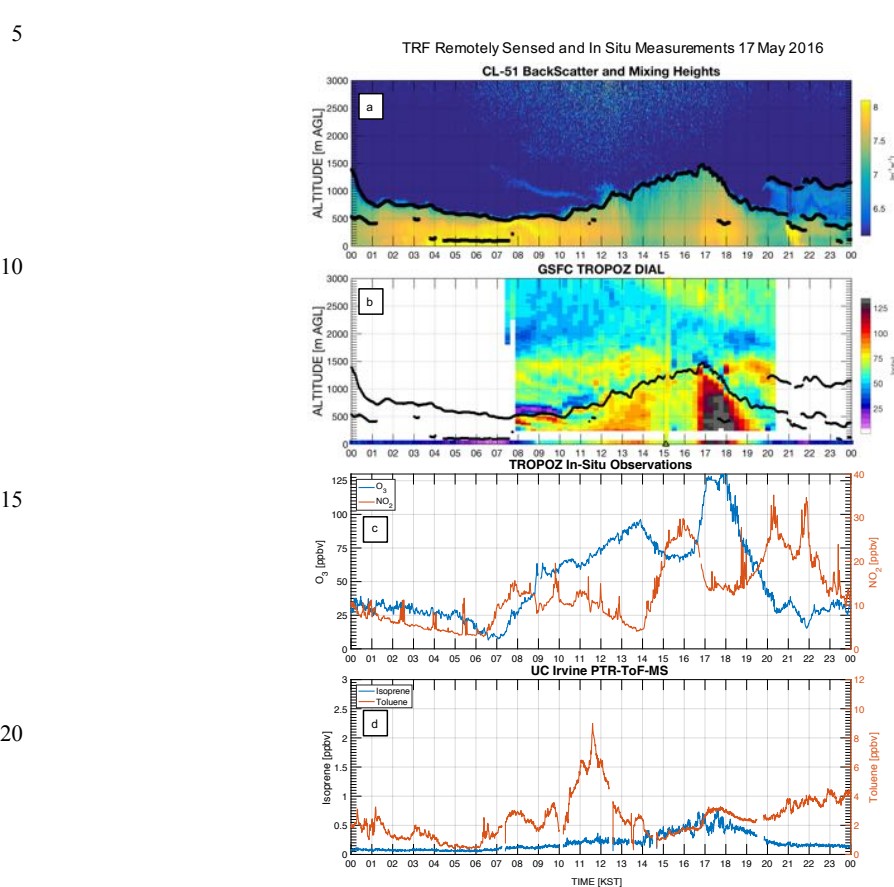

**Figure 7.** Time series of a) ceilometer backscatter profiles and mixing heights, b) GSFC TROPOZ DIAL $O_3$ profiles, c) in situ

25   surface $O_3$ and $NO_2$, and d) isoprene and toluene concentrations for 17 May 2016. Note: the DC-8 overpass occurred near

15:30 KST. The co-located ozonesonde is denoted in b) with a black triangle.



**Figure 8.** NASA DC-8 measurements of a) altitude, b) O₃, c) NO₂, d) isoprene, e) toluene and f) modeled P(O₃) from the

afternoon science flight on 17 May 2016 from 15:05 to 15:40 KST.





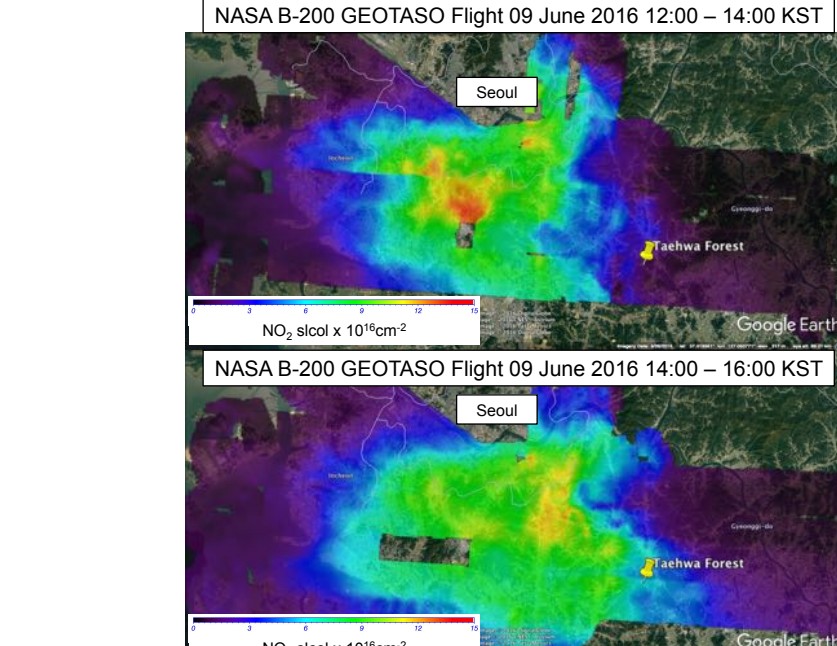

**Figure 9.** Observations of NO₂ slant columns from the GeoTASO instrument during the two afternoon science flights on 09
June 2016 from 12:00 to 14:00 and 14:00 to 16:00 KST. TRF and Seoul are denoted.



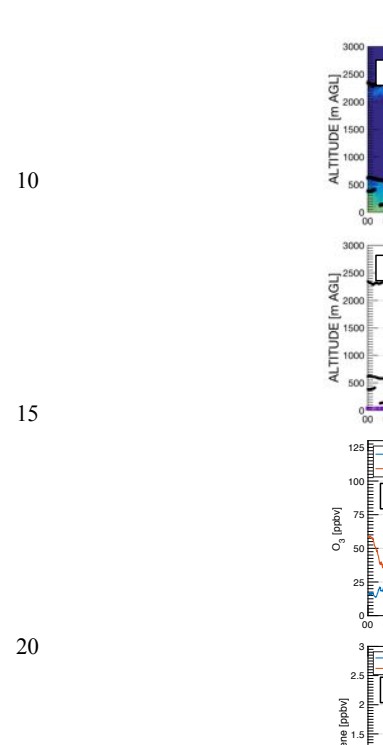

25 **Figure 10.** Time series of a) ceilometer backscatter profiles and mixing heights, b) GSFC TROPOZ DIAL $O_3$ profiles, c) in situ surface $O_3$ and $NO_2$, and d) isoprene and toluene concentrations for 09 June 2016. Note: the DC-8 overpass occurred near 15:30 KST. The co-located ozonesonde is denoted in b) with a black triangle.

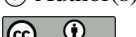



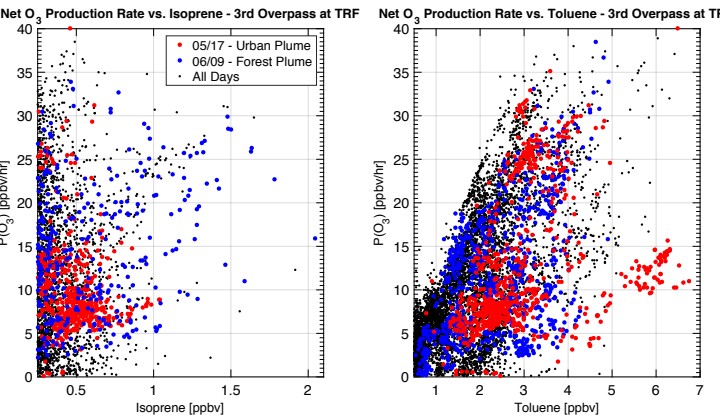

**Figure 11.** Photochemical model output of net $O_3$ production, $P(O_3)$, for all afternoon flights during the KORUS-AQ study compared to measured concentrations of isoprene (left) and toluene (right). The case studies described in Section 3 are denoted in red (17 May 2016) and blue (09 June 2016).





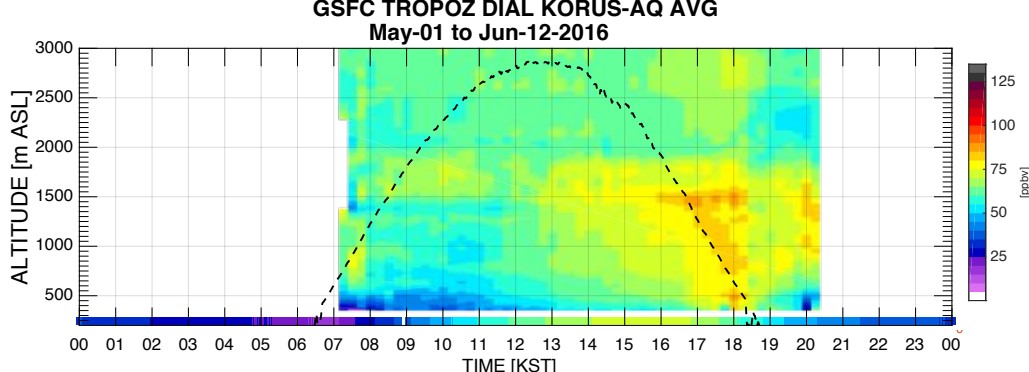

**Figure 12.** Composite averaged curtain of GSFC TROPOZ DIAL observations during the KORUS-AQ study period. This

15   comprises 31 days of lidar data and over 250 hours of measurements averaged together, and the analogous surface

measurements and solar curve proxy are also overlaid.