# Peer review of "Taehwa Research Forest: A receptor site for severe domestic pollution events in Korea during 2016"

_Atmospheric Chemistry and Physics, 2018_

## Referee Comment (RC1) · Anonymous Referee #1 · 13 Feb 2019

The manuscript by Sullivan et al. provides analysis on the atmospheric chemistry dataset taken during the 2016 KORUS-AQ campaign. This effort characterizes the chemical interaction between Seoul metropolitan area and its downwind region. The results confirm a VOC limited scenario for the ozone formation at the investigated area consistent with previous studies. This manuscript is well written and I recommend to publish after addressing following minor concerns.

Specific comments:

P19, the title of Figure 1, does the red square in (a) correspond to (b)? The letters in Figure 1 (c) and (d) are too small to see. I suggest to enlarge these two images. They don't have to sit inside (a) and (b).

[Figure]

P5, section 3.1.1, the correlation between ozone and temperature in Figure 4 doesn't look significant. Previous studies [e.g., Kuang et al 2017] suggest the correlation between ozone and temperature anomaly or water vapor anomaly especially in summertime is relatively significant from surface to free troposphere reflecting the influence of meteorological conditions on ozone photochemistry. But, if meaningful average is not available, I suggest to add a relative humidity curtain in Figure 4. In addition, I don't see a sufficient analysis on the May 17 and June 9 ozonesonde profiles while ozonesonde experts are among the co-authors. Kuang, S., Newchurch, M. J., Thompson, A. M., Stauffer, R. M., Johnson, B. J., & Wang, L. (2017). Ozone variability and anomalies observed during SENEX and SEAC4RS campaigns in 2013. Journal of Geophysical Research: Atmospheres, 122, 11,227-11,241.

P5, L27, is the surface ozone on May 17 in Figure 1 (d) much higher than the one measured by the ozonesonde shown in Figure 4? If this is true, can you comment on this difference?

Interestingly, tropospheric ozone increased a lot from 5/17 to 5/18 shown by the ozonesondes in Figure 4.

P7, L1-2, "it encounters a rapid increase in concentrations of all species, with pronounced increases in concentrations of NO2 (to 15-25 ppbv), CO (to 600-800 ppbv), and SO2 (to 4-6 ppbv)." Can you provide explanation on the extremely high CO at the end of North/land Leg, >800 ppbv, in Figure 5?

P7, L26, the 2nd "6c" should be "6e"?

P24, can you label the aircraft flying direction in Figure 6?

P8, "This is a strong indicator of an aged urban air mass containing highly reactive O3 precursors impacting rural sites." The biggest feature in Figure 6 is the enhanced ozone at the east of TRF. Do the "highly reactive O3 precursors" refer to VOC according to Figure 6?

P10, L11, replace Figure 7 with 10.

---

## Referee Comment (RC2) · Anonymous Referee #2 · 14 Feb 2019

This manuscript provided an interesting study about two air pollution episodes at the remote Taehwa Research Forest (TRF) site during the 2016 KORUS-AQ campaign. Surface observations, ozonesonde data, and aircraft measurements are used to investigate the characteristics of these two events with elevated ozone. This study suggests the outflow of air pollutants from the Seoul Metropolitan Area (SMA) plays a more important role in air pollutant at the receptor TRF site, compared with long-range transport from China. This manuscript points out the importance of domestic emissions to the local air pollution in South Korea. The topic is applicable for Atmospheric Chemistry and Physics. The text is concisely written and well documented. This study has comprehensive analysis and detailed explanation/discussion. However, the current abstract did not emphasize the major findings such as domestic emissions may have

more important impacts than the trans-boundary emissions which were believed to be the major sources of air pollution in South Korea. Re-writing the abstract is suggested. Also, lots of references are missing or not in the ACP reference format in the current manuscript, which should be added and corrected in the revised version. In summary, minor revisions as indicated in the comments and remarks below are needed before consideration of publication in ACP.

Detailed Remarks/Suggestions for Revision

Page 2 Line 6 & 7: Kim et al. (2012) and Ghim et al. (2007) are missing. Line 9: The citation should be "Duncan et al. (2016)". Please use "()" for ACP format. Line 11: Wang et al. 2015 is missing. Line 20: My understanding is that Olympic Park is also in the urban area of Seoul. Why list this reference here?

Page 3 Line 3: What is the road transport source? What is the difference than "mobile/vehicular emissions"? Line 5: Should be 'Herman et al., (2018)'. Line 7: Please add the definition of DU, i.e. "1 DU = 2.69 x 1016 molecules/cm2" here. Line 8: The statement of "O3 formation within the SMA is VOC limited, due to the overabundance of NOx" needs further discussion/explanation, such as the quantification of VOCs and NOx ratio observed in SMA urban area. Line 10: Should be "Miyazaki et al. (2018)" Line 12: Should "Kim et al., 2018" be "Kim et al., 2018a"? Line 16: Move "parts-per-billion by volume" to the first time appearance of "ppbv" Line 19-24: Using section number in "()" is hard to follow. Please use the format such as "(Section 3.1.1)".

Page 5 Line 3: Kim et al. (2014) is missing. Generally, the BVOCs emissions are elevated with higher temperature, but it depends on the plant type and stress information such as soil moisture. Just curious if Kim et al. (2014) discussed the characteristics of BVOCs emissions in South Korea.

Page 6 Line 6: Should "Lee et al., 2007" be "Lee et al., 2008"? Line 18: Huang et al. (2012) is not in the bibliography, should it be "Huang et al., (2018)"? Line 30-31: This conclusion may not be correct. During the "eastbound leg", the King Air aircraft

stayed at the same altitude of 1500 m, while the "south leg" observed air pollutants from 500-1500m over the Yellow Sea. So the "clean air" observed in "eastbound leg" may be caused by the higher altitude of King Air which could be higher than the shallow marine PBL height where the transport happens. The lower altitude like 500 m over the Yellow Sea under "eastbound leg" could have elevated concentrations of air pollutants which are not observed by King Air.

Page 7 Line 15: Should be "Figure 6" instead of "Figure 7" here. Line 24: Should "Kim et al., (2014)" be "Kim et al., (2015)"? Line 29: It is an important conclusion. Does King Air or DC-8 have NOy measurements to support this hypothesis?

Page 8 Line 22: Should "Kim et al., 2018" be "Kim et al. 2018b" in the bibliography? Line 32-33: Is it possible that due to high concentration of OH, the BVOCs concentrations are in the equilibrium and produce lots of O3, i.e., Ox?

Page 10 Line 11: "Figure 7" should be "Figure 10".

Page 11 Line 15: Does 'contributed more' mean that the biogenic emissions contribute more than anthropogenic emissions in O3 photochemical production? Since the NOx concentrations are different in these two days, some discussion may be needed here.

Page 12 Line 14-15: Kim et al. (2004) and Ghim et al. (2007) are missing in the bibliography. Line 22: Liao et al. (2019) is missing. Line 27: "Cooper et al., 2016" should be "Cooper et al., (2014)".

Page 18 Table 1: "Thompson et al., 2019" is missing.

Page 19-20 Figure 1&2: The inserted plots are too small and hard to read. May consider using individual figures.

Page 24&26 Figure 6 & 8: Numbers in color bar is blurry to read.

Page 29 Figure 11: The black, red, and blue dots covered each other. May consider using separated plots into a 3-panel figure.

---

## Author Comment (AC1) · 1 Mar 2019

Attached are the 'Author Response to Reviewers' and 'Tracked Changes of the Revised Manuscript'.

Please also note the supplement to this comment:
https://www.atmos-chem-phys-discuss.net/acp-2018-1328/acp-2018-1328-AC1-supplement.zip

---

## Author Response (AR1)

The manuscript by Sullivan et al. provides analysis on the atmospheric chemistry dataset taken during the 2016 KORUS-AQ campaign. This effort characterizes the chemical interaction between Seoul metropolitan area and its downwind region. The results confirm a VOC limited scenario for the ozone formation at the investigated area consistent with previous studies. This manuscript is well written and I recommend to publish after addressing following minor concerns.

Specific comments:

P19, the title of Figure 1, does the red square in (a) correspond to (b)? The letters in Figure 1 (c) and (d) are too small to see. I suggest to enlarge these two images. They don't have to sit inside (a) and (b).

**AR:** Figure 1 and its associated text have been restructured to clarify this. Yes, the red square inset in (a) is the spatial content of (b).

P5, section 3.1.1, the correlation between ozone and temperature in Figure 4 doesn't look significant. Previous studies [e.g., Kuang et al 2017] suggest the correlation be- tween ozone and temperature anomaly or water vapor anomaly especially in summer- time is relatively significant from surface to free troposphere reflecting the influence of meteorological conditions on ozone photochemistry. But, if meaningful average is not available, I suggest to add a relative humidity curtain in Figure 4. In addition, I don't see a sufficient analysis on the May 17 and June 9 ozonesonde profiles while ozonesonde experts are among the co-authors. Kuang, S., Newchurch, M. J., Thompson, A. M., Stauffer, R. M., Johnson, B. J., & Wang, L. (2017). Ozone variability and anomalies observed during SENEX and SEAC4RS campaigns in 2013. Journal of Geophysical Research: Atmospheres, 122, 11,227-11,241.

**AR:** The aim of the ozonesonde time series in this context is to show the overall variability in both ozone and temperature during the campaign time period. A more rigorous analysis of free tropospheric ozone concentrations and their correlations to relative humidity and/or temperature anomalies may not be as robust with only a 1 month time series. This could be a focus of future work. We have added an additional box and whisker plot for

further statistics. The box and whisker plot indicates the rarity of measuring less than 60 ppbv of $O_3$ (the 8-hour Korean standard) in the free-troposphere. We have also cited the Kuang et al paper.

P5, L27, is the surface ozone on May 17 in Figure 1 (d) much higher than the one measured by the ozonesonde shown in Figure 4? If this is true, can you comment on this difference? Interestingly, tropospheric ozone increased a lot from 5/17 to 5/18 shown by the ozonesondes in Figure 4.

**AR:** It's important to note here the timing of the balloon launch on 5/17. The launch was just after 15 LT on 5/17, before the peak pollution plume reached TRF at 17 LT. We astutely corrected for this during the field measurements on the following day and released the 5/18 sonde near 17 LT in order to capture the more likely peak in pollution. The surface monitors observed peak hourly o3 near ~130 ppbv on 5/17, but only ~90 ppbv on 5/18, as shown in Figure 1d (which has been revised from previous comments).

The ozonesonde compared well with the lidar and surface measurements during both case studies, which is illustrated in both Figures 7b and 10b.

P7, L1-2, "it encounters a rapid increase in concentrations of all species, with pronounced increases in concentrations of NO2 (to 15-25 ppbv), CO (to 600-800 ppbv), and SO2 (to 4-6 ppbv)." Can you provide explanation on the extremely high CO at the end of North/land Leg, >800 ppbv, in Figure 5?

AR: These increased CO values are attributed to localized concentrations when the aircraft was below 350 m agl and entered the vicinity of the Osan Air Base runway, where vehicle and aircraft emissions are quite high. To avoid misrepresentation of the case, we have shortened the time series from by 8 minutes (ending on 17-May-2016 11:35 LT instead of 11:43 LT).

P7, L26, the 2nd "6c" should be "6e"?

P24, can you label the aircraft flying direction in Figure 6?

AR: Arrows have been added to the flying direction in Figure 6 and Fig. 8.

P8, "This is a strong indicator of an aged urban air mass containing highly reactive O3 precursors impacting rural sites." The biggest feature in Figure 6 is the enhanced ozone at the east of TRF. Do the "highly reactive O3 precursors" refer to VOC according to Figure 6?

AR: Yes, and the reactivity is further evidenced with the increased levels of P(O3). Although toluene and isoprene are used as the tracers for VOC and BVOC estimations, the P(O3) calculations are using the entire VOC samples from the DC-8.

P10, L11, replace Figure 7 with 10.
This manuscript provided an interesting study about two air pollution episodes at the remote Taehwa Research Forest (TRF) site during the 2016 KORUS-AQ campaign. Surface observations, ozonesonde data, and aircraft measurements are used to investigate the characteristics of these two events with elevated ozone. This study suggests the outflow of air pollutants from the Seoul Metropolitan Area (SMA) plays a more important role in air pollutant at the receptor TRF site, compared with long-range trans- port from China. This manuscript points out the importance of domestic emissions to the local air pollution in South Korea. The topic is applicable for Atmospheric Chemistry and Physics. The text is concisely written and well documented. This study has comprehensive analysis and detailed explanation/discussion.

However, the current abstract did not emphasize the major findings such as domestic emissions may have more important impacts than the trans-boundary emissions which were believed to be the major sources of air pollution in South Korea. Re-writing the abstract is suggested.

**AR:** The abstract has been edited to reflect the importance of domestic emissions.

Also, lots of references are missing or not in the ACP reference format in the current manuscript, which should be added and corrected in the revised version. In summary, minor revisions as indicated in the comments and remarks below are needed before consideration of publication in ACP.

AR: Many references have been updated per referees' suggestions. We appreciate these comprehensive recommendations.

Detailed Remarks/Suggestions for Revision

Page 2 Line 6 & 7: Kim et al. (2012) and Ghim et al. (2007) are missing. Line 9: The citation should be "Duncan et al. (2016)". Please use "()" for ACP format. Line 11: Wang et al. 2015 is missing. Line 20: My understanding is that Olympic Park is also in the urban area of Seoul. Why list this reference here?

AR:  References have been corrected or removed as needed.

Page 3 Line 3: What is the road transport source? What is the difference than "mobile/vehicular emissions"? Line 5: Should be 'Herman et al., (2018)'. Line 7: Please add the definition of DU, i.e. "1 DU = 2.69 x 1016 molecules/cm2" here. Line 8: The statement of "O3 formation within the SMA is VOC limited, due to the overabundance of NOx" needs further discussion/explanation, such as the quantification of VOCs and NOx ratio observed in SMA urban area

AR: From Kim et al., 2011: Mobile/vehicular Road transport are trucks, passenger cars, taxis, light duty vehicles, buses, and RVs. Other Road Transport are Military, aircraft, railway, ships, agricultural equipment, and construction machinery. A shortened version of this list has been

added to the manuscript. More details regarding SMA being a VOC-limited region can be found in Jeon et al., (2012; 2014) and references have been added to revised manuscript.

Line 10: Should be "Miyazaki et al. (2018)" Line 12: Should "Kim et al., 2018" be "Kim et al., 2018a"?

Line 16: Move "parts- per-billion by volume" to the first time appearance of "ppbv" Line 19-24: Using section number in "()" is hard to follow. Please use the format such as "(Section 3.1.1)". Page 5 Line 3: Kim et al. (2014) is missing. Generally, the BVOCs emissions are elevated with higher temperature, but it depends on the plant type and stress information such as soil moisture. Just curious if Kim et al. (2014) discussed the characteristics of BVOCs emissions in South Korea.

AR: We apologize as this was should have been correctly cited as Kim et al., 2015 (an older ACPD version was 2014). There are several details within this manuscript (and from Lim et al., (2011)) regarding the emission rates, forest types, and speciation/composition of BVOCs through South Korea and at TRF.

Page 6 Line 6: Should "Lee et al., 2007" be "Lee et al., 2008"? Line 18: Huang et al. (2012) is not in the bibliography, should it be "Huang et al., (2018)"?

AR: Yes, this should be Lee et al., 2008 and Huang et al., 2018.

Line 30-31: This conclusion may not be correct. During the "eastbound leg", the King Air aircraft stayed at the same altitude of 1500 m, while the "south leg" observed air pollutants from 500-1500m over the Yellow Sea. So the "clean air" observed in "eastbound leg" may be caused by the higher altitude of King Air which could be higher than the shallow marine PBL height where the transport happens. The lower altitude like 500 m over the Yellow Sea under "eastbound leg" could have elevated concentrations of air pollutants which are not observed by King Air.

AR: The direct comparison of pollutants during the different aircraft measurements has been removed from the revised manuscript.

Page 7 Line 15: Should be "Figure 6" instead of "Figure 7" here. Line 24: Should "Kim et al., (2014)" be "Kim et al., (2015)"? Line 29: It is an important conclusion. Does King Air or DC-8 have NOy measurements to support this hypothesis?

AR: To better illustrate total reactive nitrogen (NOy), the below plot shows NOy, NOx, and ratio of NOx/NOy during the flight. The SMA overpass occurs with the stark jump in all species, for example a 40-50 ppbv increase in NOx near 15:19 KST. The ratio indicates the reactive nitrogen species are predominantly NOx, ranging from 70-90%, during the SMA approach. This indicates the additional reactive organic/inorganic nitrates account for a small (10-30%) percentage of total NOy, therefore the NOy budget is dominated by NOx emissions, which we infer as local pollution based on its short lifetime. A statement about this has been worked into the revised manuscript on P7 L30-35.

[Figure]

Page 8 Line 22: Should "Kim et al., 2018" be "Kim et al. 2018b" in the bibliography? Line 32-33: Is it possible that due to high concentration of OH, the BVOCs concentrations are in the equilibrium and produce lots of O3, i.e., Ox?

AR: Near 17 LT, TRF experiences a clear jump in $O_3$ by 60 ppbv and an anti-correlated decrease in $NO_2$ by 15 ppbv, (i.e. total Ox increases by 45 ppbv). Figure 7a/b show there were no changes in cloud coverage during this time and BVOC oxidation would have likely been declining with reducing insolation during this time. Rapid changes due to large chemical gradients such as these, particularly those late in the day, are predominantly linked to pollution transport of aged urban emissions. Due to the large perturbation in NOx sampled within the SMA with multiple instruments on this day, it is likely the reservoir of high $O_3$ (and increase in total Ox) is associated with advected aged emissions passing by TRF (c.f. back trajectories in Fig. 2). There are also no strong indicators in the diurnal cycle of isoprene on this day (Figure 7d) to suggest substantial BVOC emissions or subsequent oxidation.

Page 10 Line 11: "Figure 7" should be "Figure 10".

Page 11 Line 15: Does 'contributed more' mean that the biogenic emissions contribute more than anthropogenic emissions in O3 photochemical production? Since the NOx concentrations are different in these two days, some discussion may be needed here.

AR: Based on the 1-D box modeling results, isoprene contributed more to ozone production on 5/17 than isoprene contributed to ozone production on 6/9. As the reviewer points out, it is challenging to argue the relative contribution of biogenic vs anthropogenic emissions on ozone production as both days had quite different meteorology and regional emission loadings.

Page 12 Line 14-15: Kim et al. (2004) and Ghim et al. (2007) are missing in the bibliography. Line 22: Liao et al. (2019) is missing. Line 27: "Cooper et al., 2016" should be "Cooper et al., (2014)". Page 18 Table 1: "Thompson et al., 2019" is missing.

Page 19-20 Figure 1&2: The inserted plots are too small and hard to read. May consider using individual figures.

AR: This has been edited in the final manuscript and was also recommended by Reviewer #1.

Page 24&26 Figure 6 & 8: Numbers in color bar is blurry to read.

AR: A compressed figure version was submitted to meet the size limits. A final version has been prepared with higher resolution (and still within size limits).

Page 29 Figure 11: The black, red, and blue dots covered each other. May consider using separated plots into a 3-panel figure.

AR: Since the analyses regarding Figure 11 are meant to illustrate groupings of data sets, we have decided to keep this figure in its current format. It is also helpful to directly compare mixing ratios of VOCs between the two case studies, while simultaneously against the entire campaign.

[revised manuscript text omitted]